# DISSECTING ADAPTIVE METHODS IN GANS

## ABSTRACT

Adaptive methods are a crucial component widely used for training generative adversarial networks (GANs). While there has been some work to pinpoint the "marginal value of adaptive methods" in standard tasks, it remains unclear why they are still critical for GAN training. In this paper, we formally study how adaptive methods help train GANs; inspired by the grafting method proposed in Agarwal et al. (2020), we separate the magnitude and direction components of the Adam updates, and graft them to the direction and magnitude of SGDA updates respectively. By considering an update rule with the magnitude of the Adam update and the normalized direction of SGD, we empirically show that the adaptive magnitude of Adam is key for GAN training. This motivates us to have a closer look at the class of normalized stochastic gradient descent ascent (nSGDA) methods in the context of GAN training. We propose a synthetic theoretical framework to compare the performance of nSGDA and SGDA for GAN training with neural networks. We prove that in that setting, GANs trained with nSGDA recover all the modes of the true distribution, whereas the same networks trained with SGDA (and any learning rate configuration) suffer from mode collapse. The critical insight in our analysis is that normalizing the gradients forces the discriminator and generator to be updated at the same pace. We also experimentally show that for several datasets, Adam's performance can be recovered with nSGDA methods.

## 1 INTRODUCTION

Adaptive algorithms have become a key component in training modern neural network architectures in various deep learning tasks. Minimization problems that arise in natural language processing (Vaswani et al., 2017), fMRI (Zbontar et al., 2018), or min-max problems such as generative adversarial networks (GANs) (Goodfellow et al., 2014) almost exclusively use adaptive methods, and it has been empirically observed that Adam (Kingma & Ba, 2014) yields a solution with better generalization than stochastic gradient descent (SGD) in such problems (Choi et al., 2019). Several works have attempted to explain this phenomenon in the minimization setting. Common explanations are that adaptive methods train faster (Zhou et al., 2018), escape flat "saddle-point"–like plateaus faster (Orvieto et al., 2021), or handle heavy-tailed stochastic gradients better (Zhang et al., 2019; Gorbunov et al., 2022). However, much less is known about why adaptive methods are so critical for solving min-max problems such as GANs.

Several previous works attribute this performance to the superior convergence speed of adaptive methods. For instance, Liu et al. (2019) show that an adaptive variant of Optimistic Gradient Descent (Daskalakis et al., 2017) converges faster than stochastic gradient descent ascent (SGDA) for a class of non-convex, non-concave min-max problems. However, contrary to the minimization setting, convergence to a stationary point is not required to obtain satisfactory GAN performance. Mescheder et al. (2018) empirically shows that popular architectures such as Wasserstein GANs (WGANs) (Arjovsky et al., 2017) do not always converge, and yet they produce realistic images. We support this observation with our own experiments in Section 2. Our findings motivate the central question in this paper: *what factors of Adam contribute to better quality solutions over SGDA when training GANs?*

In this paper, we investigate why GANs trained with adaptive methods outperform those trained using SGDA. Directly analyzing Adam is challenging due to the highly non-linear nature of its gradient oracle and its path-dependent update rule. Inspired by the grafting approach in (Agarwal et al., 2020), we disentangle the adaptive magnitude and direction of Adam and show evidence that an algorithm using the adaptive magnitude of Adam and the direction of SGDA (which we call Ada-nSGDA) recovers the performance of Adam in GANs. Our contributions are as follows:

- In Section 2, we present the Ada-nSGDA algorithm. We empirically show that the adaptive magnitude in Ada-nSGDA stays within a constant range and does not heavily fluctuate which motivates the focus on normalized SGDA (nSGDA) which only contains the direction of SGDA.
- In Section 3, we prove that for a synthetic dataset consisting of two modes, a model trained with SGDA suffers from *mode collapse* (producing only a single type of output), while a model trained with nSGDA does not. This provides an explanation for why GANs trained with nSGDA outperform those trained with SGDA.
- In Section 4, we empirically confirm that nSGDA mostly recovers the performance of Ada-nSGDA when using different GAN architectures on a wide range of datasets.

Our key theoretical insight is that when using SGDA and any step-size configuration, either the generator $G$ or discriminator $D$ updates much faster than the other. By normalizing the gradients as done in nSGDA, $D$ and $G$ are forced to update at the same speed throughout training. The consequence is that whenever $D$ learns a mode of the distribution, $G$ learns it right after, which makes both of them learn all the modes of the distribution ~~separately~~ at the same pace.

## 1.1 RELATED WORK

**Adaptive methods in games optimization.** Several works designed adaptive algorithms and analyzed their convergence to show their benefits relative to SGDA e.g. in variational inequality problems, Gasnikov et al. (2019); Antonakopoulos et al. (2019); Bach & Levy (2019); Antonakopoulos et al. (2020); Liu et al. (2019); Barazandeh et al. (2021). Heusel et al. (2017) show that Adam locally converges to a Nash equilibrium in the regime where the step-size of the discriminator is much larger than the one of the generator. Our work differs as we do not focus on the convergence properties of Adam, but rather on the fit of the trained model to the *true* (and not empirical) data distribution.

**Statistical results in GANs.** Early works studied whether GANs memorize the training data or actually learn the distribution (Arora et al., 2017; Liang, 2017; Feizi et al., 2017; Zhang et al., 2017; Arora et al., 2018; Bai et al., 2018; Dumoulin et al., 2016). Some works explained GAN performance through the lens of optimization. Lei et al. (2020); Balaji et al. (2021) show that GANs trained with SGDA converge to a global saddle point when the generator is one-layer neural network and the discriminator is a specific quadratic/linear function. Our contribution differs as i) we construct a setting where SGDA converges to a locally optimal min-max equilibrium but still suffers from mode collapse, and ii) we have a more challenging setting since we need at least a degree-3 discriminator to learn the distribution, which is discussed in Section 3.

**Normalized gradient descent.** Introduced by Nesterov (1984), normalized gradient descent has been widely used in minimization problems. Normalizing the gradient remedies the issue of iterates being stuck in flat regions such as spurious local minima or saddle points (Hazan et al., 2015; Levy, 2016). Normalized gradient descent methods outperforms their non-normalized counterparts in multi-agent coordination (Cortés, 2006) and deep learning tasks (Cutkosky & Mehta, 2020). Our work considers the min-max setting and shows that nSGDA outperforms SGDA as it forces discriminator and generator to update at same rate.

## 1.2 BACKGROUND

**Generative adversarial networks.** Given a training set sampled from some target distribution $\mathcal{D}$, a GAN learns to generate new data from this distribution. The architecture is comprised of two networks: a generator that maps points in the latent space $\mathcal{D}_z$ to samples of the desired distribution, and a discriminator which evaluates these samples by comparing them to samples from $\mathcal{D}$. More formally, the generator is a mapping $G_{\mathcal{V}} \colon \mathbb{R}^k \to \mathbb{R}^d$ and the discriminator is a mapping $D_{\mathcal{W}} \colon \mathbb{R}^d \to \mathbb{R}$, where $\mathcal{V}$ and $\mathcal{W}$ are their corresponding parameter sets. To find the optimal parameters of these two networks, one must solve a min-max optimization problem of the form

$$\min_{\mathcal{V}} \max_{\mathcal{W}} \mathbb{E}_{X \sim p_{data}}[\log(D_{\mathcal{W}}(X))] + \mathbb{E}_{z \sim p_z}[\log(1 - D_{\mathcal{W}}(G_{\mathcal{V}}(z)))] := f(\mathcal{V}, \mathcal{W}), \quad \text{(GAN)}$$

where $p_{data}$ is the distribution of the training set, $p_z$ the latent distribution, $G_{\mathcal{V}}$ the generator and $D_{\mathcal{W}}$ the discriminator. Contrary to minimization problems where convergence to a local minimum is *required* for high generalization, we empirically verify that most of the well-performing GANs do not converge to a stationary point.

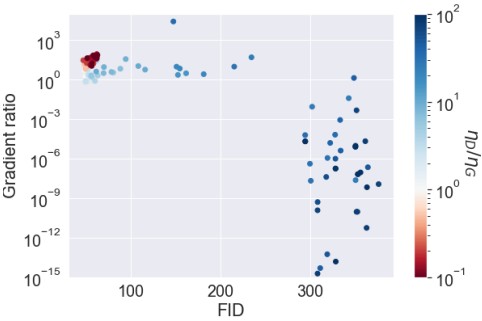

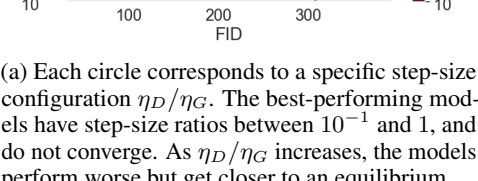

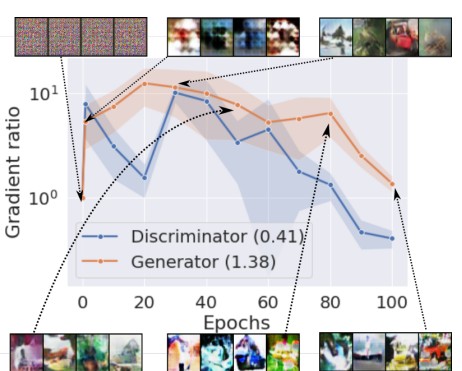

(a) Each circle corresponds to a specific step-size configuration $\eta_D/\eta_G$. The best-performing models have step-size ratios between $10^{-1}$ and $1$, and do not converge. As $\eta_D/\eta_G$ increases, the models perform worse but get closer to an equilibrium.

(b) shows that during training, the gradient ratio of a well-performing GAN approximately stays constant to 1. We also display the images produced by the model during training.

Figure 1: Gradient ratio against FID score (a) and number of epochs (b) obtained with DCGAN on CIFAR-10. This ratio is equal to $\|\text{grad}_G^{(t)}\|_2/\|\text{grad}_G^{(0)}\|_2 + \|\text{grad}_D^{(t)}\|_2/\|\text{grad}_D^{(0)}\|_2$, where $\text{grad}_G^{(t)}$ (resp. $\text{grad}_D^{(t)}$) and $\text{grad}_G^{(0)}$ (resp. $\text{grad}_D^{(0)}$) are the current and initial gradients of $G$ (resp. $D$). Note that $\|\cdot\|_2$ refers to the sum of all the parameters norm in a network. For all the plots, the models are trained for 100 epochs using a batch-size 64. For (b), the results are averaged over 5 seeds.

**Convergence and performance are decorrelated in GANs.** We support this observation through the following experiment. We train a DCGAN (Radford et al., 2015) using Adam and set up the step-sizes for $G$ and $D$ as $\eta_D, \eta_G$, respectively. Note that $D$ is usually trained faster than $G$ i.e. $\eta_D \geq \eta_G$. Figure 1a displays the GAN convergence measured by the ratio of gradient norms and the GAN's performance measured in FID score (Heusel et al., 2017). We observe that when $\eta_D/\eta_G$ is close to 1, the algorithm does not converge, but the model produces high-quality samples. On the other hand, when $\eta_D/\eta_G \gg 1$, the model converges to an equilibrium; a similar statement has been proved by Jin et al. (2020) and Fiez & Ratliff (2020) in the case of SGDA. However, the trained GAN produces low-quality solutions at this equilibrium, so simply comparing the convergence speed of adaptive methods and SGDA cannot explain the performance obtained with adaptive methods.

**SGDA and adaptive methods.** The most simple algorithm to solve the min-max (GAN) is SGDA, which is defined as follows:

$$\mathcal{W}^{(t+1)} = \mathcal{W}^{(t)} + \eta_D \mathbf{M}_{\mathcal{W},1}^{(t)}, \quad \mathcal{V}^{(t+1)} = \mathcal{V}^{(t)} - \eta_G \mathbf{M}_{\mathcal{V},1}^{(t)}, \tag{1}$$

where $\mathbf{M}_{\mathcal{W},1}^{(t)}, \mathbf{M}_{\mathcal{V},1}^{(t)}$ are the first-order momentum gradients as defined in Algorithm 1. While this method has been used in the first GANs (Radford et al., 2015), most modern GANs are trained with adaptive methods such as Adam (Kingma & Ba, 2014). The definition of this algorithm for game optimizations is given in Algorithm 1. The hyperparameters $\beta_1, \beta_2 \in [0, 1)$ control the weighting of the exponential moving average of the first and second-order moments. In many deep-learning tasks, practitioners have found that setting $\beta_2 = 0.9$ works for most problem settings. Additionally, it has been empirically observed that having no momentum (i.e., $\beta_1 \approx 0$) is optimal for many popular GAN architectures (Karras et al., 2020; Brock et al., 2018). Therefore, we only consider the case where $\beta_1 = 0$.

Optimizers such as Adam (Algorithm 1) are *adaptive* since they use a step-size for each parameter that is different than the magnitude of the gradient $\mathbf{g}_{\mathcal{Y}}^{(t)}$ for that parameter up to some constant (such as the global learning rate), and this step-size updates while training the model. There are three components that makes the adaptive update differ from the standard SGDA update: 1) the *adaptive normalization* by $\|\mathbf{g}_{\mathcal{Y}}^{(t)}\|_2$, 2) the *change of direction* from $\mathbf{g}_{\mathcal{Y}}^{(t)}/\|\mathbf{g}_{\mathcal{Y}}^{(t)}\|_2$ to $\mathbf{A}_{\mathcal{Y}}^{(t)}/\|\mathbf{A}_{\mathcal{Y}}^{(t)}\|_2$ and 3) *adaptive scaling* by $\|\mathbf{A}_{\mathcal{Y}}^{(t)}\|_2$. In summary, the steps from the standard to the adaptive update, are:

$$\mathbf{g}_{\mathcal{Y}}^{(t)} \xrightarrow[\times 1/\|\mathbf{g}_{\mathcal{Y}}^{(t)}\|_2]{\text{normalization}} \mathbf{g}_{\mathcal{Y}}^{(t)}/\|\mathbf{g}_{\mathcal{Y}}^{(t)}\|_2 \xrightarrow{\text{change of direction}} \mathbf{A}_{\mathcal{Y}}^{(t)}/\|\mathbf{A}_{\mathcal{Y}}^{(t)}\|_2 \xrightarrow[\times\|\mathbf{A}_{\mathcal{Y}}^{(t)}\|_2]{\text{adaptive scaling}} \mathbf{A}_{\mathcal{Y}}^{(t)} \tag{2}$$

The three components are entangled and it remains unclear how they contribute to the superior performance of adaptive methods relative to SGDA in GANs.

---

**Algorithm 1** Adam (Kingma & Ba, 2014) for games. All operations on vectors are element-wise.

---

**Input**: initial points $\mathcal{W}^{(0)}, \mathcal{V}^{(0)}$, step-size schedules $\{(\eta_G^{(t)}, \eta_D^{(t)})\}$, hyperparameters $\{\beta_1, \beta_2, \varepsilon\}$.

Initialize $\mathbf{M}_{\mathcal{W},1}^{(0)}, \mathbf{M}_{\mathcal{W},2}^{(0)}, \mathbf{M}_{\mathcal{V},1}^{(0)}$ and $\mathbf{M}_{\mathcal{V},2}^{(0)}$ to zero.

**for** $t = 0 \ldots T - 1$ **do**

   Receive stochastic gradients $\mathbf{g}_{\mathcal{W}}^{(t)}, \mathbf{g}_{\mathcal{V}}^{(t)}$ evaluated at $\mathcal{W}^{(t)}$ and $\mathcal{V}^{(t)}$.

   Update for $\mathcal{Y} \in \{\mathcal{W}, \mathcal{V}\}$: $\mathbf{M}_{\mathcal{Y},1}^{(t+1)} = \beta_1 \mathbf{M}_{\mathcal{Y},1}^{(t)} + \mathbf{g}_{\mathcal{Y}}^{(t)}$ and $\mathbf{M}_{\mathcal{Y},2}^{(t+1)} = \beta_2 \mathbf{M}_{\mathcal{Y},2}^{(t)} + \mathbf{g}_{\mathcal{Y}}^{(t)2}$.

   Compute gradient oracles for $Y \in \{V, W\}$: $\mathbf{A}_{\mathcal{Y}}^{(t+1)} = \mathrm{M}_{\mathcal{Y},1}^{(t+1)} / \sqrt{\mathrm{M}_{\mathcal{Y},2}^{(t+1)} + \varepsilon}$.

   Update: $\mathcal{W}^{(t+1)} = \mathcal{W}^{(t)} + \eta_D^{(t)} \mathbf{A}_{\mathcal{W}}^{(t+1)}, \qquad \mathcal{V}^{(t+1)} = \mathcal{V}^{(t)} - \eta_G^{(t)} \mathbf{A}_{\mathcal{V}}^{(t+1)}$.

---

## 2   NSGDA AS A MODEL TO ANALYZE ADAM IN GANS

In this section, we show that normalized stochastic gradient descent-ascent (nSGDA) is a suitable proxy to study the learning dynamics of Adam.

To decouple the normalization, change of direction, and adaptive scaling in Adam, we adopt the grafting approach proposed by Agarwal et al. (2020). At each iteration, we compute stochastic gradients, pass them to two optimizers $\mathcal{A}_1, \mathcal{A}_2$ and make a grafted step that combines the *magnitude* of $\mathcal{A}_1$'s step and *direction* of $\mathcal{A}_2$'s step. We focus on the optimizer defined by grafting the Adam magnitude onto the SGDA direction, which corresponds to omitting the *change of direction* in (2):

$$\mathcal{W}^{(t+1)} = \mathcal{W}^{(t)} + \eta_D \|\mathbf{A}_{\mathcal{W}}^{(t)}\|_2 \frac{\mathbf{g}_{\mathcal{W}}^{(t)}}{\|\mathbf{g}_{\mathcal{W}}^{(t)}\|_2 + \varepsilon}, \quad \mathcal{V}^{(t+1)} = \mathcal{V}^{(t)} - \eta_G \|\mathbf{A}_{\mathcal{V}}^{(t)}\|_2 \frac{\mathbf{g}_{\mathcal{V}}^{(t)}}{\|\mathbf{g}_{\mathcal{V}}^{(t)}\|_2 + \varepsilon}, \tag{3}$$

where $\mathbf{A}_{\mathcal{V}}^{(t)}, \mathbf{A}_{\mathcal{W}}^{(t)}$ are the Adam gradient oracles as in Algorithm 1 and $\boldsymbol{g}_{\mathcal{V}}^{(t)}, \boldsymbol{g}_{\mathcal{W}}^{(t)}$ the stochastic gradients. We refer to this algorithm as *Ada-nSGDA* (combining the Adam magnitude and SGDA direction). There are two natural implementations for nSDGA. In the *layer-wise* version, $\mathcal{Y}^{(t)}$ is a single parameter group (typically a layer in a neural network), and the updates are applied to each group. In the *global* version, $\mathcal{Y}^{(t)}$ contains all of the model's weights.

In Fig. 2a, we see that Ada-nSGDA and Adam appear to have similar learning dynamics in terms of the FID score. Both Adam and Ada-nSGDA significantly outperform SGDA as well as AdaDir, which is the alternate case of (3) where we instead graft the magnitude of the SGDA update to the direction of the Adam update. AdaDir diverged after a single step so we omit it in Fig. 2. These results show that the *adaptive scaling* and *normalization* components are sufficient to recover the performance of Adam, suggesting that Ada-nSGDA is a valid proxy for Adam

A natural question that arises is how the total adaptive magnitude varies during training. We empirically investigate this by tracking the layer-wise adaptive magnitudes of the Adam gradient oracles when training a GAN with Ada-nSGDA, and summarize our key findings here (see Section 4 for complete experimental details). We first train a WGAN-GP (Arjovsky et al., 2017) model, and find that the adaptive magnitude is bound within a constant range, and that all the layers have approximately the same adaptive magnitude (Fig. 2 (b,c)). This suggests that the *adaptive scaling* component is constant (in expectation) and motivates the use of *nSGDA*, corresponding to Ada-nSGDA with a constant *adaptive scaling* factor. We then train a WGAN-GP model with nSGDA and we find that nSGDA mostly recovers the FID score obtained by Ada-nSGDA (Fig. 2a).

We also validate this observation for more complicated GAN architectures by repeating this study on StyleGAN2 (Karras et al., 2019). We find that the adaptive magnitudes also vary within a constant range, but each layer has its own constant scaling factor. Thus, training StyleGAN2 with nSGDA and a global normalization fails, but training with nSGDA with a different constant step-size for each layer yields a performance that mostly recovers that of Ada-nSGDA (Fig 5). These results suggest that the schedule of the *adaptive scaling* is not central in the success of Ada-nSGDA in GANs. Instead, adaptive methods are successful because they *normalize* the gradients for each layer, which allows for more balanced updates between $G$ and $D$ as we will show in Section 3. We conduct more experiments in Section 4 and in Appendix A.

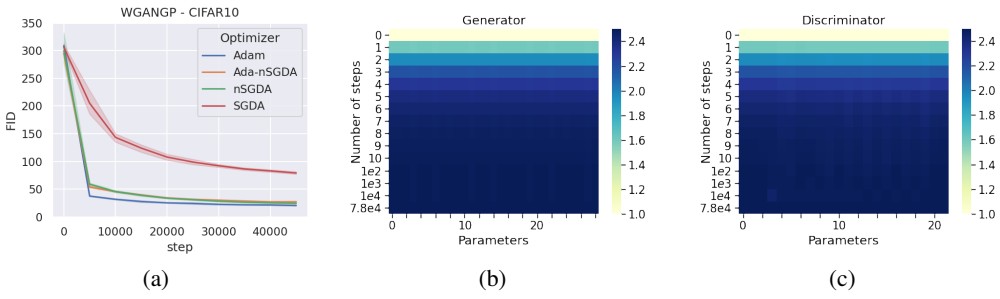

Figure 2: (a) shows the FID training curve for a WGAN-GP ResNet, averaged over 5 seeds. We see that Ada-nSGDA and nSGDA have very similar performance to Adam for a WGAN-GP. (b, c) displays the fluctuations of Ada-nSGDA adaptive magnitude. We plot the ratio $\|\mathbf{A}_{\mathcal{Y}}^{(t)}\|_2/\|\mathbf{A}_{\mathcal{Y}}^{(0)}\|_2$ for each of the generator's (b) and discriminator's (c) layers. At early stages, this ratio barely increases and remains constant after 10 steps.

## 3 WHY DOES NSGDA PERFORM BETTER THAN SGDA IN GANS?

In Section 2, we empirically showed that the most important component of the adaptive magnitude is the *normalization*, and that nSGDA (an algorithm consisting of this component alone) is sufficient to recover most of the performance of Ada-nSGDA (and by extension, Adam). Our goal is to construct a dataset and model where we can prove that a model trained with nSGDA generates samples from the true training distribution while SGDA fails. To this end, we consider a dataset where the underlying distribution consists of two modes, defined as vectors $u_1, u_2$, that are slightly correlated (see Assumption 1) and consider the standard GANs' training objective. We show that a GAN trained with SGDA using any reasonable[1] step-size configuration suffers from *mode collapse* (Theorem 3.1); it only outputs samples from a single mode which is a weighted average of $u_1$ and $u_2$. Conversely, nSGDA-trained GANs learn the two modes separately (Theorem 3.2).

**Notation** We set the GAN 1-sample loss $L_{\mathcal{V},\mathcal{W}}^{(t)}(X, z) = \log(D_{\mathcal{W}}^{(t)}(X)) + \log(1 - D_{\mathcal{W}}^{(t)}(G_{\mathcal{V}}^{(t)}(z)))$. We denote $\mathbf{g}_{\mathcal{Y}}^{(t)} = \nabla_{\mathcal{Y}} L_{\mathcal{V},\mathcal{W}}^{(t)}(X, z)$ as the 1-sample stochastic gradient. We use the asymptotic complexity notations when defining the different constants e.g. $\mathrm{poly}(d)$ refers to any polynomial in the dimension $d$, $\mathrm{polylog}(d)$ to any polynomial in $\log(d)$, and $o(1)$ to a constant $\ll d$. We denote $a \propto b$ for vectors $a$ and $b$ in $\mathbb{R}^d$ if there is a positive scaling factor $c > 0$ such that $\|a - cb\|_2 = o(\|b\|_2)$.

### 3.1 SETTING

In this section, we present the setting to sketch our main results in Theorem 3.1 and Theorem 3.2. We first define the distributions for the training set and latent samples, and specify our GAN model and the algorithms we analyze to solve (GAN). Note that for many assumptions and theorems below, we present informal statements which are sufficient to capture the main insights. The precise statements can be found in Appendix B.

Our synthetic theoretical framework considers a bimodal data distribution with two correlated modes:

**Assumption 1** ($p_{data}$ structure). *Let $\gamma = \frac{1}{\mathrm{polylog}(d)}$. We assume that the modes are correlated. This means that $\langle u_1, u_2 \rangle = \gamma > 0$ and the generated data point $X$ is either $X = u_1$ or $X = u_2$.*

Next, we define the latent distribution $p_z$ that $G_{\mathcal{V}}$ samples from and maps to $p_{data}$. Each sample from $p_z$ consists of a data-point $z$ that is a binary-valued vector $z \in \{0, 1\}^{m_G}$, where $m_G$ is the number of neurons in $G_{\mathcal{V}}$, and has non-zero support, i.e. $\|z\|_0 \geq 1$. Although the typical choice of a latent distributions in GANs is either Gaussian or uniform, we choose $p_z$ to be a binary distribution because it models the weights' distribution of a hidden layer of a deep generator; Allen-Zhu & Li (2021) argue that the distributions of these hidden layers are sparse, non-negative, and non-positively correlated. We now make the following assumptions on the coefficients of $z$:

**Assumption 2** ($p_z$ structure). *Let $z \sim p_z$. We assume that with probability $1 - o(1)$, there is only one non-zero entry in $z$. The probability that the entry $i \in [m_G]$ is non-zero is $\Pr[z_i = 1] = \Theta(1/m_G)$.*

In Assumption 2, the output of $G_{\mathcal{V}}$ is only made of one mode with probability $1 - o(1)$. This avoids summing two of the generator's neurons, which may cause mode collapse.

---

[1]Reasonable simply means that the learning rates are bounded to prevent the training from diverging.

To learn the target distribution $p_{data}$, we use a linear generator $G_{\mathcal{V}}$ with $m_G$ neurons and a non-linear neural network with $m_D$ neurons:

$$G_{\mathcal{V}}(z) = Vz = \sum_{i=1}^{m_G} v_i z_i\,, \qquad D_{\mathcal{W}}(X) = \text{sigmoid}\Big( a \sum_{i=1}^{m_D} \langle w_i, X \rangle^3 + \frac{b}{\sqrt{d}} \Big). \qquad (4)$$

where $V = [v_1^\top, v_2^\top, \cdots, v_{m_G}^\top] \in \mathbb{R}^{m_G \times d}$, $z \in \{0,1\}^{m_G}$, $W = [w_1^\top, \ldots, w_{m_D}^\top] \in \mathbb{R}^{m_D \times d}$, and $a, b \in \mathbb{R}$. Intuitively, $G_{\mathcal{V}}$ outputs linear combinations of the modes $v_i$. We choose a cubic activation as it is the smallest monomial degree for the discriminator's non-linearity that is sufficient for the generator to recover the modes $u_1, u_2$.[2]

We now state the SGDA and nSGDA algorithms used to solve the GAN training problem (GAN). For simplicity, we set the batch-size to 1. The resultant update rules for SGDA and nSGDA are:[3]

SGDA: at each step $t > 0$, sample $X \sim p_{data}$ and $z \sim p_z$ and update as

$$\mathcal{W}^{(t+1)} = \mathcal{W}^{(t)} + \eta_D \mathbf{g}_{\mathcal{W}}^{(t)}, \quad \mathcal{V}^{(t+1)} = \mathcal{V}^{(t)} - \eta_G \mathbf{g}_{\mathcal{V}}^{(t)}, \qquad (5)$$

nSGDA: at each step $t > 0$, sample $X \sim p_{data}$ and $z \sim p_z$ and update as

$$\mathcal{W}^{(t+1)} = \mathcal{W}^{(t)} + \eta_D \frac{\mathbf{g}_{\mathcal{W}}^{(t)}}{\|\mathbf{g}_{\mathcal{W}}^{(t)}\|_2}, \quad \mathcal{V}^{(t+1)} = \mathcal{V}^{(t)} - \eta_G \frac{\mathbf{g}_{\mathcal{V}}^{(t)}}{\|\mathbf{g}_{\mathcal{V}}^{(t)}\|_2}. \qquad (6)$$

Compared to the versions of SGDA and Ada-nSGDA that we introduced in Section 2, we have the same algorithms except that we set $\beta_1 = 0$ and omit $\varepsilon$ in (5) and (6). Note that since there is only one layer in the neural networks we study in this paper, the global-wise and layer-wise versions of nSGDA are actually the same. Lastly, we detail how to set the optimization parameters for SGDA and nSGDA in (5) and (6).

**Parametrization 3.1** (Informal). *When running SGDA and nSGDA on (GAN), we set:*

  – ***Initialization***: $b^{(0)} = 0$, and $a^{(0)}$, $w_i^{(0)} (i \in [m_D])$, $v_j^{(0)} (j \in [m_G])$ are initialized with a Gaussian with small variance.

  – ***Number of iterations***: we run SGDA for $t \leq T_0$ iterations where $T_0$ is the first iteration such that the algorithm converges to an approximate first order local minimum. For nSGDA, we run for $T_1 = \tilde{\Theta}(1/\eta_D)$ iterations.

  – ***Step-sizes***: For SGDA, $\eta_D, \eta_G \in (0, \frac{1}{\text{poly}(d)})$ can be arbitrary. For nSGDA, $\eta_D \in (0, \frac{1}{\text{poly}(d)}]$, and $\eta_G$ is slightly smaller than $\eta_D$.

  – ***Over-parametrization***: For SGDA, $m_D, m_G = \text{polylog}(d)$ are arbitrarily chosen i.e. $m_D$ may be larger than $m_G$ or the opposite. For nSGDA, we set $m_D = \log(d)$ and $m_G = 2\log(d)$.

Our theorem holds when running SGDA for any (polynomially) possible number of iterations; after $T_0$ steps, the gradient becomes inverse polynomially small and SGDA essentially stops updating the parameters. Additionally, our setting allows any step-size configuration for SGDA i.e. larger, smaller, or equal step-size for $D$ compared to $G$. Note that our choice of step-sizes for nSGDA is the one used in practice, i.e. $\eta_D$ slightly larger than $\eta_G$.

## 3.2 MAIN RESULTS

We state our main results on the performance of models trained using SGDA (5) and nSGDA (6). We show that nSGDA learns the modes of the distribution $p_{data}$ while SGDA does not.

**Theorem 3.1** (Informal). *Consider a training dataset and a latent distribution as described above and let Assumption 1 and Assumption 2 hold. Let $T_0$, $\eta_G, \eta_D$ and the initialization be as defined in*

---

[2]Li & Dou (2020) show that when using linear or quadratic activations, the generator can fool the discriminator by only matching the first and second moments of $p_{data}$.

[3]In the nSGDA algorithm defined in (3), the step-sizes were time-dependent. Here, we assume for simplicity that the step-sizes $\eta_D, \eta_G > 0$ are *constant*.

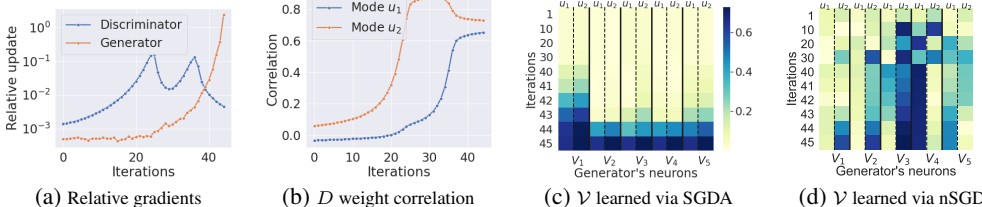

(a) Relative gradients     (b) $D$ weight correlation     (c) $\mathcal{V}$ learned via SGDA     (d) $\mathcal{V}$ learned via nSGDA

Figure 3: (a) shows the relative gradient updates for SGDA. $D$ first updates its weights while $G$ does not move until iteration 20, then $G$ moves its weights. (b) shows the correlation for one neuron of $D$ (with maximal correlation to $u_2$ at initialization) with the modes $u_1, u_2$ during the learning process of SGDA. (c, d) shows the correlations of the neurons of $G$ with the modes when trained with SGDA and nSGDA respectively. This shows that for SGDA (c), the model ultimately learns the weighted average $u_1 + u_2$. For nSGDA, we see from (d) that one of the neurons ($V_4$) is highly correlated with $u_1$ and another one ($V_3$) is correlated with $u_2$.

*Parametrization 3.1. Let $t$ be such that $t \leq T_0$. Run SGDA on (GAN) for $t$ iterations with step-sizes $\eta_G, \eta_D$. Then, with probability at least $1 - o(1)$, the generator outputs for all $z \in \{0,1\}^{m_G}$:*

$$G_{\mathcal{V}}^{(t)}(z) \propto \begin{cases} u_1 + u_2 & \text{if } \eta_D \geq \eta_G, \\ \xi^{(t)}(z) & \text{otherwise} \end{cases}, \tag{7}$$

*where $\xi^{(t)}(z) \in \mathbb{R}^d$ is some vector that is not correlated to any of the modes. Formally, $\forall \ell \in [2]$, $\cos(\xi^{(t)}(z), u_\ell) = o(1)$ for all $z \in \{0,1\}^{m_G}$.*

A formal proof can be found in Appendix G. Theorem 3.1 indicates that when training with SGDA and any step-size configuration, the generator either does not learn the modes at all ($G_{\mathcal{V}}^{(t)}(z) = \xi^{(t)}(z)$) or learns an average of the modes ($G_{\mathcal{V}}^{(t)}(z) \propto u_1 + u_2$). The theorem holds *for any* time $t \leq T_0$ which is the iteration where SGDA converges to an approximate first-order locally optimal min-max equilibrium. Conversely, nSGDA succeeds in learning the two modes separately:

**Theorem 3.2** (Informal). *Consider a training dataset and a latent distribution as described above and let Assumption 1 and Assumption 2 hold. Let $T_1$, $\eta_G, \eta_D$ and the initialization as defined in Parametrization 3.1. Run nSGDA on (GAN) for $T_1$ iterations with step-sizes $\eta_G, \eta_D$. Then, the generator learns both modes $u_1, u_2$ i.e., for $\ell \in \{1, 2\}$,*

$$\Pr_{z \sim p_z}[G_{\mathcal{V}}^{(T_1)}(z) \propto u_\ell] \quad \text{is non-negligible}. \tag{8}$$

A formal proof can be found in Appendix I. Theorem 3.2 indicates that when we train a GAN with nSGDA in the regime where the discriminator updates slightly faster than the generator (as done in practice), the generator successfully learns the distribution containing the direction of both modes.

We implement the setting introduced in Subsection 3.1 and validate Theorem 3.1 and Theorem 3.2 in Fig. 3. Fig. 3a displays the relative update speed $\eta \|\mathbf{g}_{\mathcal{Y}}^{(t)}\|_2 / \|\mathcal{Y}^{(t)}\|_2$, where $\mathcal{Y}$ corresponds to the parameters of either $D$ or $G$. Fig. 3b shows the correlation $\langle w_i^{(t)}, u_\ell \rangle / \|w_i^{(t)}\|_2$ between *one* of $D$'s neurons and a mode $u_\ell$ and Fig. 3c the correlation $\langle v_j^{(t)}, u_\ell \rangle / \|v_j^{(t)}\|_2$ between $G$'s neurons and $u_\ell$. We discuss the interpretation of these plots to the next section.

### WHY DOES SGDA SUFFER FROM MODE COLLAPSE AND NSGDA LEARN THE MODES?

We now explain why SGDA suffers from mode collapse, which corresponds to the case where $\eta_D \geq \eta_G$. Our explanation relies on the interpretation of Figs. 3a, 3b, and 3c, and on the updates around initialization that are defined as followed. There exists $i \in [m_D]$ such that $D$'s update is

$$\mathbb{E}[w_i^{(t+1)} | w_i^{(t)}] \approx w_i^{(t)} + \eta_D \sum_{l=1}^{2} \mathbb{E}[\langle w_i^{(t)}, u_l \rangle^2] u_l. \tag{9}$$

Thus, the weights of $D$ receive gradients directed by $u_1$ and $u_2$. On the other hand, the weights of $G$ at early stages receive gradients directed by $w_j^{(t)}$:

$$v_i^{(t+1)} \approx v_i^{(t)} + \eta_G \sum_j \langle v_i^{(t)}, w_j^{(t)} \rangle^2 w_j^{(t)}. \tag{10}$$

We observe that the learning process in Figs. 3a & 3b has three distinct phases. In the first phase (iterations 1-20), $D$ learns one of the modes ($u_1$ or $u_2$) of $p_{data}$ (Fig. 3b) and $G$ barely updates its weights (Fig. 3a). In the second phase (iterations 20-40), $D$ learns the weighted average $u_1 + u_2$ (Fig. 3b) while $G$ starts moving its weights (Fig. 3a). In the final phase (iterations 40+), $G$ learns $u_1 + u_2$ (Fig. 3c) from $D$. In more detail, the learning process is described as follows:

**Phase 1** : At initialization, $w_j^{(0)}$ and $v_j^{(0)}$ are small. Assume w.l.o.g. that $\langle w_i^{(0)}, u_2 \rangle > \langle w_i^{(0)}, u_1 \rangle$. Because of the $\langle w_i^{(t)}, u_l \rangle^2$ in front of $u_2$ in (9), the parameter $w_i^{(t)}$ gradually grows its correlation with $u_2$ (Fig. 3b) and $D$'s gradient norm thus increases (Fig. 3a). While $\|w_j^{(t)}\| \ll 1 \,\forall j$, we have that $v_i^{(t)} \approx v_i^{(0)}$ (Fig. 3a).

**Phase 2**: $D$ has learned $u_2$. Because of the sigmoid in the gradient of $w_i^{(t)}$ (that was negligible during Phase 1) and $\langle u_1, u_2 \rangle = \gamma > 0$, $w_i^{(t)}$ now mainly receives updates with direction $u_2$. Since $G$ did not update its weights yet, the min-max problem (GAN) is approximately just a minimization problem with respect to $D$'s parameters. Since the optimum of such a problem is the weighted average $u_1 + u_2$, $w_j^{(t)}$ slowly converges to this optimum. Meanwhile, $v_i^{(t)}$ start to receive some significant signal (Fig. 3a) but mainly learn the direction $u_1 + u_2$ (Fig. 3c), because $w_j^{(t)}$ is aligning with this direction.

**Phase 3:** The parameters of $G$ only receive gradient directed by $u_1 + u_2$. The norm of its relative updates stay large and $D$ only changes its last layer terms (slope $a$ and bias $b$).

In contrast to SGDA, nSGDA ensures that $G$ and $D$ always learn at the same speed with the updates:

$$w_i^{(t+1)} \approx w_i^{(t)} + \eta_D \frac{\langle w_i^{(t)}, X \rangle^2 X}{\| \langle w_i^{(t)}, X \rangle^2 X \|_2}, \text{ and } v_i^{(t+1)} \approx v_i^{(t)} + \eta_G \frac{\sum_j \langle w_j^{(t)}, v_i^{(t)} \rangle^2 w_j^{(t)}}{\| \sum_j \langle w_j^{(t)}, v_i^{(t)} \rangle^2 w_j^{(t)} \|_2} \quad (11)$$

No matter how large $\langle w_i^{(t)}, X \rangle$ is, $G$ still learns at the same speed with $D$. There is a tight window (iteration 25, Fig. 3b) where only one neuron of $D$ is aligned with $u_1$. This is when $G$ can also learn to generate $u_1$ by "catching up" to $D$ at that point, which avoids mode collapse.

## 4 NUMERICAL PERFORMANCE OF NSGDA

In Section 2, we presented the Ada-nSGDA algorithm (3) which corresponds to "grafting" the Adam magnitude onto the SGDA direction. In Section 3, we construct a dataset and GAN model where we prove that a GAN trained with nSGDA can generate examples from the true training distribution, while a GAN trained with SGDA fails due to mode collapse. We now provide more experiments comparing nSGDA and Ada-nSGDA with Adam on real GANs and datasets.

We train a ResNet WGAN with gradient penalty on CIFAR-10 (Krizhevsky et al., 2009) and STL-10 (Coates et al., 2011) with Adam, Ada-nSDGA, SGDA, and nSGDA with a fixed learning rate as done in Section 3. We use the default architectures and training parameters specified in Gulrajani et al. (2017) ($\lambda_{GP} = 10$, $n_{dis} = 5$, learning rate decayed linearly to 0 over 100k steps). We also train a StyleGAN2 model (Karras et al., 2020) on FFHQ (Karras et al., 2019) and LSUN Churches (Yu et al., 2016) (both resized to $128 \times 128$ pixels) with Adam, Ada-nSGDA, SGDA, and nSGDA. We use the recommended StyleGAN2 hyperparameter configuration for this resolution (batch size = 32, $\gamma = 0.1024$, map depth = 2, channel multiplier = 16384). We use the Fréchet Inception distance (FID) (Heusel et al., 2017) to quantitatively assess the performance of the model. For each optimizer, we conduct a coarse log-space sweep over step sizes and optimize for FID. We train the WGAN-GP models for 2880 thousand images (kimgs) on CIFAR-10 and STL-10 (45k steps with a batch size of 64), and the StyleGAN2 models for 2600 kimgs on FFHQ and LSUN Churches. We average our results over 5 seeds for the WGAN-GP ResNets, and over 3 seeds for the StyleGAN2 models due to the computational cost associated with training GANs.

**WGAN-GP** Figures 4a and 4b validates the conclusions on WGAN-GP from Section 2. We find that both Ada-nSGDA and nSGDA mostly recover the performance of Adam, with nSGDA obtaining a final FID of ∼2-3 points lower than Ada-nSGDA. As discussed in Section 2, such performance is possible because the adaptive magnitude stays within a constant range. In contrast, models trained with SGDA consistently perform significantly worse, with final FID scores $4\times$ larger than Adam.

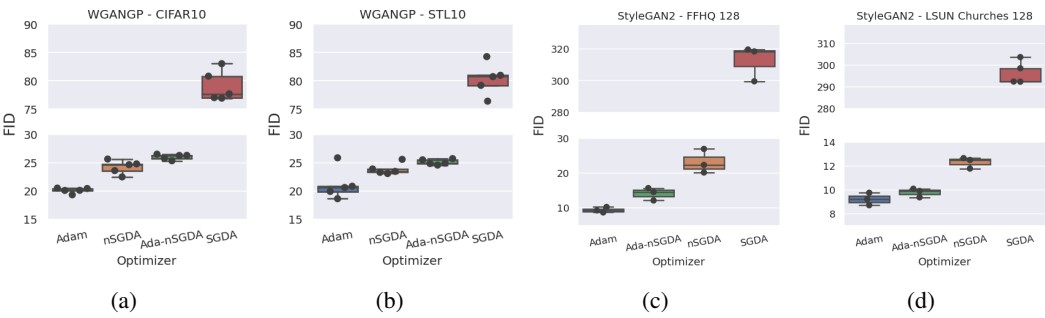

(a)          (b)          (c)          (d)

Figure 4: (a, b) are the final FID scores (5 seeds) for a ResNet WGAN-GP model trained for 45k steps on CIFAR-10 and STL-10 respectively. (c, d) are the final FID scores (3 seeds) for a StyleGAN2 model trained for 2600kimgs on FFHQ and LSUN Churches respectively. We use the same constant layer scaling in (d) for nSGDA as that in (c), which was found by tracking the layer-wise adaptive step-sizes.

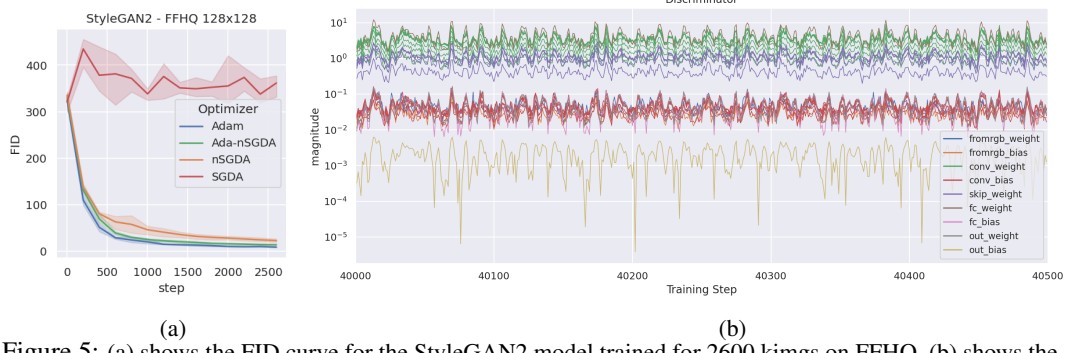

(a)          (b)

Figure 5: (a) shows the FID curve for the StyleGAN2 model trained for 2600 kimgs on FFHQ. (b) shows the fluctuations of the Ada-nSGDA adaptive magnitude for each layer over an arbitrary slice of 500 training steps for the Discriminator. The layers are grouped by common types, e.g. Conv weights and biases, etc.). We find that although the magnitude for each layer fluctuates, the fluctuations are bounded to some fixed range for each layer. We show similar behaviour for the Generator in the Appendix.

**StyleGAN2**    Figures 4c and 4d show the final FID scores when training a StyleGAN2. We find that Ada-nSGDA recovers most of the performance of Adam, but one difference with WGAN-GP is that nSGDA does not work if we use the same global learning rate for each layer. As discussed in Section 2, nSGDA with a different (but constant) step-size for each layer *does* work, and is able to mostly recover Ada-nSGDA's performance (Fig. 5a). To choose the scaling for each layer, we train StyleGAN2 with Ada-nSGDA on FFHQ-128, track the layer-wise adaptive magnitudes, and take the mean of these magnitudes over the training run (for each layer). Figure 5b shows that the fluctuations for each layer are bound to a constant range, validating our assumption of constant step-sizes. Additionally, the same scaling obtained from training FFHQ seems to work for different datasets; we used it to train StyleGAN2 with nSGDA on LSUN Churches-128 and recovered similar performance to training on this dataset with Ada-nSGDA (Fig. 4d).

## 5    CONCLUSION

Our work addresses the question of which mechanisms in adaptive methods are critical for training GANs, and why they outperform non-adaptive methods. We empirically show that Ada-nSGDA, an algorithm composed of the adaptive magnitude of Adam and the direction of SGD, recovers most of the performance of Adam. We further decompose the adaptive magnitude into two components: normalization, and adaptive step-size. We then show that the adaptive step size is roughly constant (bounded fluctuations) for multiple architectures and datasets. This empirically indicates that the normalization component of the adaptive magnitude is the key mechanism of Ada-nSGDA, and motivates the study of nSGDA; we verify that it too recovers the performance of Ada-nSGDA. Having shown that nSGDA is a good proxy for a key mechanism for adaptive methods, we then construct a setting where we proved that nSGDA –thanks to its balanced updates– recovers the modes of the true distribution while SGDA fails to do it. The key insight from our theoretical analysis is that the ratio of the update of $D$ and $G$ must be close to 1 during training in order to recover the modes of the distribution. This matches the experimental setting with nSGDA, as we find that global norm of the parameter updates for both $D$ and $G$ are almost equal for optimal choices of learning rates.

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
