# OpenReview forum: "Dissecting adaptive methods in GANs"
_ICLR.cc/2023/Conference — Submitted to ICLR 2023_

### Official Review · Reviewer_GLnC · 2022-10-24

**Confidence:** 3
**Correctness:** 3
**Technical Novelty And Significance:** 4
**Empirical Novelty And Significance:** 4
**Recommendation:** 8

**Clarity, Quality, Novelty And Reproducibility:**

The paper is clear, but it would improve readability if the conclusion would come back on some contributions. Especially, the proof about mode-collapse not necessarily being related to convergence, and the importance of the range of the gradients seems important in the paper and are not mentioned.
The work is original in the way it frames its question and in the used methods to answer it.


**Strength And Weaknesses:**

Strengths:
- The paper tries to answer a challenging question and shed lights on some standard practices with GANs. A better understanding the training dynamics of GANs is essential for improving the current state-of-the-art, and this work brings some valuable insights in that regard.
- The analysis encompasses both theoretical insights and solid empirical evidence, including results with StyleGAN2 that diverge from the theoretical assumptions.

Weaknesses (and questions):
- To the question "Why does nSGDA perform better than SGDA in GANs?" (Section 3) the authors answer by building a scenario where they can prove that nSGDA succeeds while SGDA fails. The demonstration is compelling but not entirely rigorous as it seems likely one could very well build a scenario where the opposite happens.
- For the StyleGAN-2 plots in Figure 4c and 4d, while the authors mention that "nSGDA was not able to recover the performance of Adam", the actual numbers still seem relevant to the discussion. Can the authors provide them and add them to the figure?
- As mentioned by the authors, StyleGAN-2 is vastly different from the other GANs, which makes the interpretation of the results difficult. It could be insightful to apply the analysis to BigGAN [1] or FastGAN [2] which might be closer to the theoretical framework.
- The authors suggest multiple times throughout the paper that what is important is that the ratio of the magnitudes stays in range. This should be discussed in a condensed form in the conclusion. Also, could it become an actionable diagnosis tool for monitoring GAN training?

---

[1] Andrew Brock, Jeff Donahue, Karen Simonyan. Large Scale GAN Training for High Fidelity Natural Image Synthesis. ICLR 2019

[2] Bingchen Liu, Yizhe Zhu, Kunpeng Song, Ahmed Elgammal. Towards Faster and Stabilized GAN Training for High-fidelity Few-shot Image Synthesis. ICLR 2021.

**Summary Of The Paper:**

The paper investigates why there is a huge difference in performance between the Adam optimizer and the SGDA optimizer for training GANs.
To do so, it proposes to consider separately, in Adam, the effects of the adaptive magnitude of gradients and those of the direction of the gradients. Therefore, the authors conduct experiments with normalized SGDA (nSGDA) and introduce variants by combining either the direction of nSGDA and the adaptive magnitudes of Adam (Ada-nSGDA), or the direction of Adam and the magnitudes of SGDA (AdaDir).
They build a synthetic setting on which they develop proof that nSGDA recovers the target distribution while SGDA cannot, and use this experiment to draw some insight into the training dynamics.
Finally, they show empirically that only Adam, Ada-nSGDA, and in some cases nSGDA achieve good results, and conclude that the balancing of magnitudes of updates is the key.

**Summary Of The Review:**

It was a very interesting read. To this day, the training dynamics of GANs are still obscure and cause problems (for instance on few data, or conversely, to scale to very large datasets or long training time). In that sense, this paper tackles an important question and there is a strong possibility that the findings could in the future be refined to be of practical use.

I also appreciated the inclusion of StyleGAN-2 in the empirical studies. It is important for this kind of study to check if their findings could be at least partially transferred to more modern architectures and training setups.

---

### Official Review · Reviewer_Vs2s · 2022-10-27

**Confidence:** 4
**Correctness:** 3
**Technical Novelty And Significance:** 3
**Empirical Novelty And Significance:** 2
**Recommendation:** 5

**Clarity, Quality, Novelty And Reproducibility:**

It is well written and clear to understand and follow. The authors also shared their codes.

**Strength And Weaknesses:**

Strengths:

1. They follow a systematic and theoretically sensible approach to show the importance of adaptive magnitude.


2. The proofs of the theorems seem plausible.


3. Some of their empirical results are compatible to the theory.

 Weaknesses:

1. In Figure 4(a) and (b), we should expect to see the superiority of Ada-nSGDA to nSGDA in terms of FID since the main objective of the paper is to show the necessity of the adaptive magnitude. The authors already explained this result is possible because of nSGDA’s competence. If nSGDA is already enough to recover Adam’s performance, how can one elucidate the importance of adaptive magnitude?

2. For StyleGAN2, the authors state that nSGDA cannot recover the performance of Adam. However, they did not show the FID scores of nSGDA in Figure 4(c) and (d). We should be able to visually compare nSGDA with the other three. Besides, if normalized directions are not enough in this case, can we conclude that Ada-nSGDA’s success is the result of adaptive magnitude? The authors did not discuss these results in terms of adaptive magnitude.

3. Even though the objective is to emphasize the importance of adaptive magnitude, the authors put a lot more eﬀort to prove and show the superiority of nSGDA over SGDA and its compatibility to Adam than to compare Ada-nSGDA to nSGDA. As an example, in Appendix A.2 Figure 12, it would be good to see the results of Ada-nSGDA compared with layer-wise and global nSGDAs which would also support the conclusion.


**Summary Of The Paper:**

In this paper, the authors study the underlying reasons of adaptive methods’ success in GAN training. They decouple the factors of Adam, namely adaptive magnitude and direction, and graft the adaptive magnitude to the normalized direction of SGDA to obtain Ada-nSGDA. By doing so, they manage to compare the eﬀects of magnitude and direction in a systematic way. In order to prove the suitability of normalized directions, they also show the superiority of nSGDA to SGDA in terms of mode collapse. Lastly, they present empirical results to conﬁrm Ada-nSGDA’s compatibility to Adam with regards of GAN’s performance using various GAN architectures and datasets. Taking everything into consideration, they conclude that the adaptive magnitude of Adam is the main reason for the success of adaptive methods in GANs.

**Summary Of The Review:**

In this paper, the authors investigate the eﬀects of the adaptive methods’ components which are adaptive magnitude and direction to the GAN performance. As the nature of Adam optimizer makes the disentanglement to these components hard, the authors ﬁrst present nSGDAs which have compatible performance with Adam and that they outperform SGDAs. Once they obtain this suitable method for directions, they graft the adaptive magnitude of Adam to the direction of nSGDA (Ada-nSGDA) so that they can analyze the eﬀects of magnitude and direction of adaptive methods in a systematic way. Even though the idea is valuable and the authors clearly show the outperformance of nSGDAs to SGDAs both theoretically and experimentally, I believe they should show the performance improvements of Ada-nSGDAs over nSGDAs in order to conclude that the underlying reason of adaptive methods’ success is adaptive magnitude instead of direction. Their experimental results seem lacking in this context.

---

### Official Review · Reviewer_3Jvo · 2022-11-03

**Confidence:** 3
**Correctness:** 3
**Technical Novelty And Significance:** 4
**Empirical Novelty And Significance:** 4
**Recommendation:** 5

**Clarity, Quality, Novelty And Reproducibility:**

### Clarity, quality
I think that the presentation is not clear enough. See the Weaknesses section as well as the Minor points.
### Novelty
Although I did not do extensive literature survey, the proposal seems to be novel.
### Reproducibility
The authors provide, in the supplementary material, patches to be applied to existing codebases implementing WGAN-GP and StyleGAN2 to reproduce the program codes used in their experiments, although I did not investigate them.

**Strength And Weaknesses:**

### Strength
- This paper provides an interesting working hypothesis about why and how adaptive learning methods are useful in training of GANs.
- Theoretical results were obtained for learning behaviors of SGDA and nSGDA, even though under a quite idealistic setting.
### Weaknesses
1. In the last paragraph, the authors claimed that nSGDA, which is explained as "Ada-nSGDA where we omit the adaptive magnitudes" (I guess that it is nothing other than SGDA with gradient normalized to be of unit norm), performs as well as Adam. Regarding this claim:
  - If normalizing the gradients to be of unit norm be sufficient for good performance, how can one justify the key claim in this paper that the adaptive magnitudes are essential for good performance?
  - The theoretical development in Section 3 is based on SGDA and nSGDA. One can then argue that the theoretical consideration in this paper fails to distinguish between constant-norm gradients (nSGDA) and the adaptive magnitudes (Ada-nSGDA).
  - The same criticism can also be raised against the discussion on the StyleGAN2 experiments in page 9. It might suggest, as the authors are claiming throughout this paper, importance of the adaptive magnitudes in Ada-nSGDA, but this claim is not well supported by the theoretical argument in this paper.
2. The discussion is inconsistent in many other places as well. For example:
  - On page 8, line 12, one sees reference to Fig. 3 (b) while discussing behaviors of nSGDA. Figure 3 (b) shows, however, results for SGDA, not for nSGDA, so that the reference here does not make sense.
  - On page 8, lines 34 and 44, the authors claim that Figures 4 (a)-(d) show the FID *curves during training*, whereas these figures show the final FID scores.
  - On page 8, line 46, the authors claimed that nSGDA did not perform as well as Adam. Why did not they show the results in Figure 4 (c), (d) in order to support the claim?
### Minor points
- Page 2, line 2: An extra space after the word "direction".
- Page 2, line 13: What "its counterpart" refers to is unclear.
- Page 2, line 46: The domain of the generator $G_{\mathcal{V}}$ is written as $\mathbb{R}^k$, but it should be the latent space $\mathcal{D}_z$.
- Page 2, line 46: The range of the discriminator $D_{\mathcal{W}}$ should not be $\mathbb{R}$ but $(0,1)$.
- Page 2, line 49: The operator $:=$ in equation (GAN) should be $=:$.
- Page 2, lines 49, 50: Is $p_{\mathrm{data}}$ different from $\mathcal{D}$?
- Page 3, Figure 1 caption: The definition of gradient ratio should be put in the main text.
- Page 3, Figure 1 caption: I did not understand what "all the parameters norm" means. Should not $\\|\cdot\\|_2$ be squared?
- Page 3, line 11: most simple $\to$ simplest
- Page 3, line 17: The first sentence is somehow duplicate of the statement in line 14.
- Page 7, line 13: (to $\to$ in) the next section.
- Page 7, line 17: as follow(ed $\to$ s).
- Page 7, line 29: in front of ($u_2$ $\to$ $u_l$)
- Page 9, line 1: What does "the adaptive magnitude the ratio" mean? Also, $\\|\mathbf{A}_\mathcal{Y}^{(t)}\\|_2$ is itself not a ratio.
- Page 9, line 25: the same perform(ance)

**Summary Of The Paper:**

This paper studies why and how adaptive learning methods help training of GANs. The authors postulate that the adaptive magnitudes of gradients of Adam is a key. This postulate leads them to propose, with the step-size grafting approach by Agarwal et al., two algorithms, Ada-nSGDA and nSGDA, the former combining the Adam magnitude and SGDA direction, whereas the latter omitting the Adam magnitude from Ada-nSGDA. Theoretical analysis is then done on behaviors of SGDA and nSGDA under an idealistic setting, showing that SGDA fails to learn the modes (Theorem 3.1) and that nSGDA succeeds under reasonable conditions on the learning rates (Theorem 3.2). Numerical experiments with WGAN-GP and StyleGAN2 were conducted to empirically confirm the key observation that the adaptive magnitudes is a key to success of Adam in GAN training.

**Summary Of The Review:**

Although this paper provides an interesting working hypothesis on why adaptive learning methods are helpful in training GANs, I feel that the arguments in this paper are not convincing enough, as detailed in the Weaknesses section. I would therefore evaluate this paper as being slightly negative.

---

### Official Review · Reviewer_WMSK · 2022-11-04

**Confidence:** 4
**Correctness:** 3
**Technical Novelty And Significance:** 2
**Empirical Novelty And Significance:** 2
**Recommendation:** 5

**Clarity, Quality, Novelty And Reproducibility:**

Clarity: The contents are clear, but some parts are inconclusive.

Quality:  Some claims are not well-supported by evidence. There are some minor errors.

Novelty: The conclusions are somehow well-known.

Reproducibility: Good. I didn't run the code, but I've checked the README and the patch files provided. It seems runnable.

**Strength And Weaknesses:**

Strength
- The paper is well-organized and easy to understand.
- The paper includes abundant analyses, both empirically and theoretically.
- The code is provided for reproduction.

Weakness
- The contribution is limited as it is well-known that adaptive mechanisms are useful in SGD-based optimization. Also, the claimed key insight about GAN training does not very beneficial to the community.
- The claims or conclusions are not well-supported by the empirical and theoretical studies.
    - Section 2:  As a study of optimization methods, empirical study on only one architecture and dataset is not general enough.
    - Section 3: While the authors claim adaptive magnitude is essential to Adam, the conclusions about nSGDA, which use neither magnitude nor direction from Adam, seem not really relevant to Adam.
    - Section 3: The assumptions are so restricted that the conclusion is not general. Does SGDA always mode collapse? Does nSGDA never mode collapse? Does nSGDA always better than SGDA?
- Some sections are not conclusive.
    - Section 2: The authors claim that the critical component of Adam is the adaptive magnitude because of the comparison between Ada-nSGDA and SGDA. However, if we look at the comparison between Ada-nSGDA and nSGDA, the adaptive magnitude seems not important.
    - Section 4: Similar to above, nSGDA works better than Ada-nSGDA.
- Some minor problems.
    - Section 1, 2nd paragraph: SGDA not defined yet
    - Fig 2 (b), (c) the order of generator and discriminator is not consistent with the caption.
    - Evaluated methods are described repeatedly in the 2nd and 3rd paragraphs in Section 4.

**Summary Of The Paper:**

This work investigates the Adam optimizer for GAN training. The authors analyze the optimization steps by separating the magnitude and the direction of weight updates. They consider nSGDA, a normalized version of the standard stochastic gradient descent ascent (SGDA), and Ada-nSGDA, a combination of the magnitude of Adam and the direction of nSGDA, as alternatives to analysis Adam. The paper is composed of three parts.
1. The authors use Ada-nSGDA to show that the adaptive magnitude is essential in optimization empirically.
2. A theoretical study shows that under some assumptions, nSGDA can perfectly recover the distribution but SGDA can not. Then they analyze the optimization process of GAN empirically.
3. They empirically confirm that Ada-nSGDA and nSGDA perform similarly to Adam using two GAN architectures and three datasets.

**Summary Of The Review:**

This paper works on an interesting problem of figuring out how Adam works for GAN training. The authors conduct rich empirical and theoretical studies. However, those studies are not convincing enough to make general conclusions. The contribution of this work is limited.

---

### Decision · Program_Chairs · 2023-01-20

**Decision:**

Reject

**Justification For Why Not Higher Score:**

The paper has some interesting findings, however, as shown in the experiments, the findings of the paper may be of less broad interest of ICLR audience.

**Justification For Why Not Lower Score:**

N/A

**Metareview: Summary, Strengths And Weaknesses:**

Summary:
The authors investigated the Adam optimizer for GAN training and therefore proposed nSGDA and Ada-nSGDA, which is a gradient graft of Adam and SGD. Experimental and theoretical analysis demonstrates the correct direction of the investigation.

Strength:
It is a very interesting idea to analyze the different gradient components of Adam for GAN training. The paper is well-written and well-justified.

Weakness:
The experimental results are not always supporting the theoretical analysis. There is no clear margin of nSGDA over Adam. Therefore, the contributions to the community are very limited.

**Summary Of Ac-Reviewer Meeting:**

This paper receives 1 strong positive (8 score) and 3 weak negatives (5 scores). In fact, all the reviewers appreciated the theoretical contribution of the work and were concerned about the experimental results. The positive and negative camps only disagree on whether the practice is more important than the theory or not. AC read the paper and the reviewer-author discussion and is leaning toward the negative camp because the only bright side of the theory is not convincing as the dissection of Adam is not a sufficient condition for better GAN training, not to mention the questionable experimental results. Therefore, the paper can be considered as a paper with interesting findings, which is marginally below the bar of ICLR main conference paper.